

# Musical timbre style transfer with diffusion model

Hong Huang[1], Junfeng Man[2,3], Luyao Li[1] and Rongke Zeng[1]

[1] School of Computer Science, Hunan University of Technology, Zhuzhou, China
[2] School of Intelligent Manufacturing, Hunan First Normal University, Changsha, China
[3] Key Laboratory of Industrial Equipment Intelligent Perception and Maintenance in College of Hunan Province, Hunan First Normal University, Changsha, China

## ABSTRACT

In this work, we focus on solving the problem of timbre transfer in audio samples. The goal is to transfer the source audio's timbre from one instrument to another while retaining as much of the other musical elements as possible, including loudness, pitch, and melody. While image-to-image style transfer has been used for timbre and style transfer in music recording, the current state of the findings is unsatisfactory. Current timbre transfer models frequently contain samples with unrelated waveforms that affect the quality of the generated audio. The diffusion model has excellent performance in the field of image generation and can generate high-quality images. Inspired by it, we propose a kind of timbre transfer technology based on the diffusion model. To be specific, we first convert the original audio waveform into the constant-Q transform (CQT) spectrogram and adopt image-to-image conversion technology to achieve timbre transfer. Lastly, we reconstruct the produced CQT spectrogram into an audio waveform using the DiffWave model. In both many-to-many and one-to-one timbre transfer tasks, we assessed our model. The experimental results show that compared with the baseline model, the proposed model has good performance in one-to-one and many-to-many timbre transfer tasks, which is an interesting technical progress.

## INTRODUCTION

In recent years, researchers have been paying more and more attention to the use of artificial intelligence in music creation. One important area of research in this discipline is the study of music style transfer. Before delving into this inquiry, let us suppose for the purposes of experimentation that there are two distinct aspects of music: style and content. It is possible to construct whole new compositions with unique qualities from various sources by dividing and recombining these two elements. Research on music style transfer is broadly categorized into timbre style transfer (*Lu et al., 2019*; *Cífka et al., 2021*), performance style transfer (*Xiao, Chen & Zhou, 2023*; *Mukherjee & Mulimani, 2022*), and composition style transfer (*Hung et al., 2019*; *Nakamura et al., 2019*), among other categories, with different definitions for content and style.

Timbre style transfer has garnered significant attention in the field of music style transfer. Timbre is a significant characteristic of musical sound. However, the definition of timbre is ambiguous, and there are multiple interpretations of its meaning in existing

Corresponding author
Junfeng Man, mjfok@qq.com

literature. The definition by the Acoustical Society of America describes timbre as a perceptual attribute that enables listeners to differentiate between two similar sounds that have the same loudness and pitch. For example, a person may play the same note on a violin and a piano, yet people can still discern the differences between these two instruments. This definition intuitively captures the human perception of timbre, emphasizing its uniqueness and high recognizability. Despite the fact that this definition makes it very evident what timbre is not, timbre is a very abstract concept that is challenging to define succinctly. Although sometimes compared to visual colors, timbre is more difficult to quantify than visual colors. Modeling timbre is a challenging task because different instruments exhibit great differences in the time and frequency domains (*Wessel, 1979*). While there are well-designed physical models that can simulate sound production using formulas and parameters (*Karplus & Strong, 1983*; *McIntyre, Schumacher & Woodhouse, 1983*; *Smith, 1992*), hi-fi sample libraries are still preferred in virtual instrument plugins.

Numerous techniques for creating new sounds from two or more sources are available in the literature on music signal processing. Some of these algorithms include analysis-re-synthesis (*Jehan, 2004*; *Masri, Bateman & Canagarajah, 1997*), cross-synthesis (*Burred, 2013*; *Lazzarini & Timoney, 2009*), vocoding (*Mor et al., 2018*), and hybridization (*Donin & Traube, 2016*). Regarding style transfer techniques, parallel datasets with stylistic notes in the target domain that correspond to each other in the source domain are typically needed for feature interpolation (*Caetano & Rodet, 2011*) or matrix decomposition (*Driedger, Prätzlich & Müller, 2015*; *Su et al., 2017*) based on conventional style transfer techniques. Stated differently, style transfer requires supervised element-by-element specification of content properties. The system's application is severely limited by this shortcoming. The latest advancements in deep learning models have unlocked new opportunities for analyzing high-dimensional data and managing intricate subsequent tasks. Researchers have recently put out a range of models for learning musical instrument timbre and music style transfer that are based on machine learning techniques. For example, *Brunner et al. (2018)* introduced a model called MIDI-VAE, which is based on a variational autoencoder. This model achieves polyphonic music style transfer for multiple instrument tracks. Although the genre is decomposed through the auxiliary classifier, other musical attributes are intertwined. *Huang et al. (2018)* proposed Timbretron, which utilizes Constant-Q Transform (CQT) spectrograms as the representation of the original audio waveform. They employ CycleGAN as the timbre transfer model and then use a pre-trained WaveNet to convert the transformed CQT features back into the original audio. Their method captures higher resolution at lower frequencies while maintaining pitch variance, but the synthesized audio contains unrelated artifacts, resulting in suboptimal audio quality. *Mor et al. (2018)* modeled the original waveform and achieved timbre transfer between multiple instruments by training a shared WaveNet encoder and multiple independent WaveNet decoders. However, this approach requires training multiple decoders, incurring significant computational costs. The audio samples need to be generated sequentially, which slows down the audio synthesis and makes it less suitable for

real-time applications. *Jain et al. (2020)* proposed an attention-based method for timbre transfer (ATT) that allows for the transfer of the timbre of the source audio from one instrument to another. *Chen & Chen (2021)* proposed a musical instrument timbre transfer model based on a multi-channel attention guidance mechanism.The application of a multichannel attention guidance mechanism enhances the ability of the model guidance generator to capture the most discriminative components (harmonic components) of the spectrogram. *Engel, Gu & Roberts (2019)* proposed that a differentiable digital signal processing (DDSP) model can also perform timbre transfer, but is limited to monophonic music. Although these studies have achieved good results in their respective areas, they did not propose explicit solutions for enhancing the quality of spectrogram generation. Moreover, most of their methods can only perform one-to-one transfer operations between two given instruments and cannot achieve more flexible control over timbre. In multi-instrument timbre transfer operations, previous methods have complex structures, high computational costs, and slow audio generation speed.

Note that parallel research that slightly overlaps with this work emerged during the completion of this work. *Comanducci, Antonacci & Sarti (2023)*. proposed a timbre transfer method, DiffTransfer, based on denoising diffusion implicit models (DDIMs). The work in this article has several key advances and differences compared to DiffTransfer compared to DiffTransfer, there are several key advances and differences:(1) The method in this article introduces a new cross-attention mechanism in the potential layer that allows for more accurate learning of the timbre of the target instrument from the conditioning mechanism. This enhancement can address some of the model's limitations in capturing complex timbre nuances. (2) In this article, we introduce a potential layer compression strategy that can reduce the data dimensionality more efficiently, thus speeding up the model inference. (3) The model in this article introduces a conditional mechanism that enables the execution of unpaired timbre transfer, increasing the applicability of the model.

In this article, we delve into the issue of timbre style transfer, focusing on two main aspects:(1) timbre transfer while maintaining musical content and sound quality. (2) Achieving more flexible one-to-one and many-to-many timbre transfer in audio waveforms containing only single and multiple instruments. Specifically, the possibility of shifting between audio tracks containing only a single instrument and a mix of instruments without a prior separation step is investigated. A perceived aspect of musical sound that is distinct from pitch and amplitude contours is referred to as musical timbre (*Colonel & Keene, 2020*). This work adopts a simplistic perspective, assuming an equivalency between instruments and timbres. This work aims to achieve the conversion of a musical composition from one vocal timbre type to another, while maintaining other qualities that are significant to the music. In this study, three primary task types were implemented. The first is the one-to-one transfer of a single timbre, which in this case refers to converting the timbre of one instrument to that of another while maintaining the musical content (*e.g.*, pitch, rhythm, and melody). For example, the timbre of a piano performance is converted to the timbre of a violin, while maintaining the pitch and rhythm of the original performance. The one-to-many transfer of a single timbre comes next. For example, a

trumpet solo can be changed into a solo for a flute, cello, or guitar while keeping the same melodic structure. Finally, there is many-to-many transfer of multiple timbres, where given a source audio containing multiple timbres, the goal is to replace the timbre of each sound according to a predefined mapping while preserving the content of all the sounds. Specifically, all sounds that originally had a first input timbre will be replaced with a first output timbre, all sounds with a second input timbre will be replaced with a second output timbre, and so on. For example, a piece of music played by a piano, and a violin is converted to a piece of music played by an electric guitar, and a flute, while maintaining the pitch, rhythm, and melody of each instrumental part.

This study proposes a framework for music timbre transfer based on the diffusion model (*Ho, Jain & Abbeel, 2020*) and conducts tests for both individual and multiple instruments. To achieve the transfer of timbre from one instrument to another, an intriguing strategy is to directly apply image-based style transfer techniques to the time-frequency representation of audio. Here, we utilize a similar technique for image-to-image style transfer (*Saharia et al., 2022*). We treat the music style transfer problem as a multimodal conditional distribution of learning styles in the target domain given only one unpaired sample in the source domain. Specifically, this article utilizes noise as the input for the spread model and adjusts it by selecting the target timbre CQT spectrogram chart as the input. It then gradually learns to reconstruct the desired timbre's CQT spectrogram through a denoising process. This process involves adjusting the CQT spectrogram based on the selected target timbre. However, restoring the style-transformed CQT spectrogram back to an audio waveform presents a fundamental obstacle. Accurate reconstruction requires phase information, which is difficult to predict. Existing phase inference techniques, such as *Griffin & Lim (1984)*, introduce disturbances in the output waveform, creating unnecessary artifacts that affect the quality of audio generation. Recently, deep generative models based on neural networks, such as WaveNet (*Oord et al., 2016*; *Rethage, Pons & Serra, 2018*) and WaveRNN (*Paul, Pantazis & Stylianou, 2020*; *Gu et al., 2021*), have achieved significant success in generating high-quality audio. However, these models suffer from slow inference generation speeds, which makes them inefficient for real-time usage. Therefore, this article adopts the DiffWave model (*Kong et al., 2020*), which is based on the Diffusion model, to restore the generated spectrogram to the original audio. The DiffWave model has a faster inference speed compared to other models, while still being able to generate high-quality audio. For time-frequency representation, this article utilizes the CQT method to convert the original audio waveform into a spectrogram. Unlike the short-time Fourier transform (STFT), the CQT has specific advantages in representing music audio. It better captures the fundamental frequency of notes, which increases exponentially with higher pitches. To validate the effectiveness of the proposed model, we employ both objective measurements and subjective evaluations. The experimental results demonstrate that the proposed model can successfully perform one-to-one and many-to-many timbre transfers. This is an interesting technical advance compared to the baseline model. The audio demonstrations for this article can be found at https://youtu.be/DTINzPH_LBI.

## RELATED WORK

### Timbre transfer

Previous deep learning models have addressed various aspects of music style transfer research, such as timbre style transfer, composition style transfer, and performance style transfer. In recent years, there has been a significant focus on timbre style transfer, which involves converting the timbre of one instrument in audio to that of another instrument. For music style transfer, broadly speaking, the implementation method mainly involves two aspects: the transfer can be of symbolic music or audio signals. In this article, we will focus on studying the timbre style transfer of audio signals.

*Verma & Smith (2017)* were the first to apply deep learning models to timbre transfer. Subsequent studies have suggested a model that is founded on the WaveNet autoencoder architecture (*Mor et al., 2018*) for the transformation of musical waveforms across various stylistic domains, including different musical instruments. This approach entails direct analysis of the audio waveform, whereas an alternative method involves utilizing the time-frequency representation of the audio. The study in *Jain et al. (2020)* employs an attention-based architecture to transform the Mel-spectrogram, which is subsequently inverted using the MelGAN (*Kumar et al., 2019*) architecture. Other studies employ CycleGAN-based models (*Lu et al., 2019*; *Chen & Chen, 2021*; *Yang, Cinquin & Sörös, 2021*), which eliminate the need for paired training data and integrate consistency loss terms in the reconstruction process to achieve accurate content reconstruction. Several studies have also explored variational autoencoders (*Chang, Chen & Hu, 2021*; *Bitton, Esling & Chemla-Romeu-Santos, 2019*), which demonstrate the ability to transfer timbre across diverse domains. Nevertheless, this method is restricted to monophonic musical compositions. Additionally, there exists a category of techniques that involve the extraction of musical parameters (*Engel, Gu & Roberts, 2019*) from the input track, including pitch and loudness. These parameters are then transmitted through the process of sound resynthesis and are utilized in conjunction with deep learning models to generate tracks with the desired timbre. Contemporary research predominantly depends on generation models such as CycleGAN (*McAllister & Gambäck, 2022*), VAE (*Cífka et al., 2021*), UNIT (*Liu, Breuel & Kautz, 2017*), and musicVAE (*Roberts et al., 2018*). Although these methods have contributed significantly to the examination of timbre style, the outcomes of their application suggest that certain limitations exist.

In this study, we will employ a time-frequency representation approach to investigate a flexible timbre style transfer framework that enables multi-instrument timbre transfer without any source separation preprocessing. Simultaneously, this study also places greater emphasis on the quality of the generated spectrogram in order to produce higher quality audio.

### Diffusion model

The denoising diffusion probability model (DDPM) (*Ho, Jain & Abbeel, 2020*) is widely acknowledged as a prominent model for generative tasks at the forefront of current research. Due to the simplicity and effectiveness of DDPM training and its superior

generation results, it has increasingly replaced generative adversarial networks (GAN) (*Goodfellow et al., 2020*) and variational autoencoders (VAE) (*Peng et al., 2021*) in the field of image generation. Despite the potent generative capabilities of DDPM, its sampling process is recognized for its sluggish pace. This requirement is necessary to ensure that the sampling process conforms to the Markov chain. The DDIM model, which is an extension of the DDPM, is designed to enhance the sampling process iteratively by employing non-Markov methods (DDIM; *Song, Meng & Ermon, 2020*). A recent study introduced the potential diffusion model (LDM) (*Rombach et al., 2022*) with the aim of reducing computational complexity by compressing images from the pixel space to the potential space for diffusion. Moreover, it demonstrates strong capabilities in generating images. The remarkable generation capabilities of the diffusion model render this announcement a noteworthy advancement for the field of music generation. In recent times, the diffusion model has been applied in the field of audio technology. In a study conducted by *Hawthorne et al. (2022)*, the diffusion model was effectively utilized to transform MIDI tracks into spectrograms, demonstrating its ability to modify audio representations. Another study (*Schneider et al., 2024*) introduces a model for transforming text into music, offering an innovative approach to translating written content into musical compositions. Furthermore, the diffusion model has shown significant effectiveness in various audio-related applications, such as symbolic music generation (*Mittal et al., 2021*), speech synthesis (*Kong et al., 2020*; *Chen et al., 2020*), and song extraction (*Plaja-Roglans, Miron & Serra, 2022*). The diverse applications of the diffusion model in the audio domain demonstrate its capacity to provide innovative solutions for audio processing tasks.

This article will employ the latent diffusion model for timbre style transfer due to its quicker sampling time and superior quality outcomes. In this respect, the work in this article has some commonalities with DiffTransfer (*Comanducci, Antonacci & Sarti, 2023*), as both use diffusion modeling to achieve the transfer of instrumental timbre styles. In order to improve the quality of the generated audio, the vocoder model DiffWave (*Kong et al., 2020*) based on diffuion model is used in this study. It is notably simpler to train compared to GAN-based models (*Kumar et al., 2019*) and exhibits significantly faster reasoning speed than WaveNet (*Oord et al., 2016*; *Rethage, Pons & Serra, 2018*).

## METHODS

This article presents a flexible system for transferring music timbre, with the objective of establishing a mapping from the source audio domain $X$ to the target audio domain $Y$. It is feasible to transform the timbre of an instrument from the source audio domain $X$ to that of the target instrument in the target audio domain $Y$ while preserving characteristics such as tone and loudness. The system has the capability to achieve one-to-one or many-to-many timbre style transfers. As depicted in Fig. 1, the operation of the system is primarily segmented into three components. Firstly, the input audio waveform employs CQT to acquire the spectrogram representation of music. Furthermore, the CQT spectrogram does not take into account the phase information and instead treats it as an image. We approach the timbral transfer as a problem of converting one image to another. In this study, the timbre transfer model LDM is utilized to perform the timbre transfer operation.

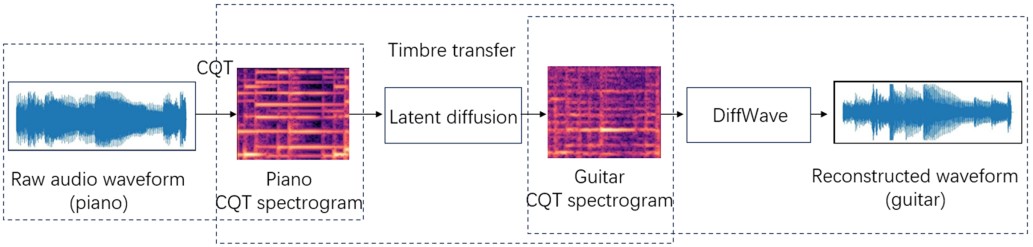

**Figure 1  Piano to guitar timbre transfer.**

Subsequently, the CQT representation produced following the transfer of timbres is converted back into the audio waveform through the utilization of the trained DiffWave model.

Subsequently, the three steps outlining the functioning of the system will be expounded upon. This section will focus on the timbre transfer.

## Time-frequency representation

Timbre is a complex and unquantifiable attribute, making it very challenging to directly train models using raw audio waveforms. This article aims to solve the problem by employing a method that transforms abstract concepts into concrete forms. It achieves style transfer from image to image by obtaining the spectrogram diagram of audio and ultimately restoring the processed spectrogram diagram to audio. In time-frequency analysis, two very representative algorithms are the short-time Fourier transform (STFT) and the CQT. STFT is one of the most commonly used time-frequency analysis methods, while CQT is considered particularly suitable for analyzing music data. Therefore, this article will utilize the CQT method to represent raw audio data.

Before introducing the CQT, it is important to briefly examine the concept of the twelve-tone equal temperament. An octave pitch is uniformly divided into 12 semitones, with the frequency ratio of the two adjacent notes being $\sqrt[12]{2}$. This suggests that the distribution of music pitch follows an exponential pattern rather than linearly. The center frequency of the CQT, an important method for time-frequency analysis, is distributed according to an exponential law. This results in varying filtering bandwidths while maintaining a constant Q ratio of center frequency to bandwidth. The main characteristic of CQT that sets it apart from other popular time-frequency analysis techniques is its frequency axis, which is scaled logarithmically instead of linearly. Additionally, the window size of CQT varies with changes in frequency. This allows the CQT to align with the distribution of scale frequencies, providing significant advantages when analyzing musical signals.

In the context of time-frequency analysis, this article partitions the audio into 5-s segments to facilitate processing. Each segmented audio clip undergoes a transformation using CQT to convert it into a spectrogram. In this procedure, the article utilizes the Hanning window with a sampling rate of 16 kHz and a step length of 256 for each column. It acquires the 84-dimensional logarithmic scale CQT representation for each frame, encompassing seven octaves, each of which comprises 12 dimensions.

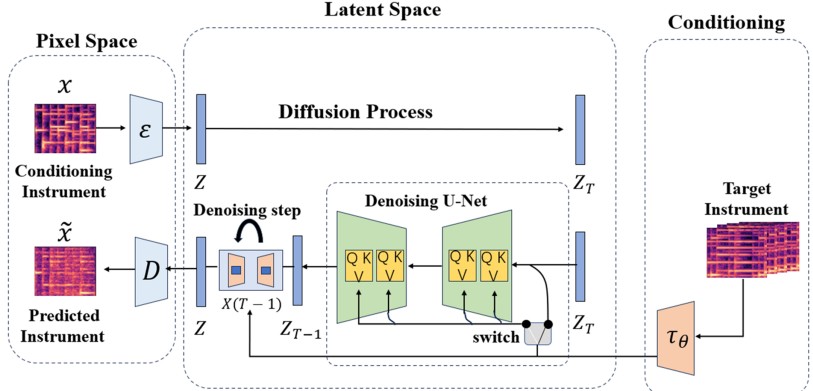

**Figure 2  Models of timbre transfer.**

## Timbre style transfer

This section will outline the proposed architecture for timbre style transfer. The architecture accepts the CQT spectrogram diagram of the source instrument as input and produces the corresponding target CQT spectrogram diagram. The CQT diagram presented is derived from the performance of the same musical piece on the specified target instrument. As depicted in Fig. 2, the timbre transfer model comprises three main components. The first model is the VAE in pixel space, designed to capture the input and output information of the CQT spectrogram for the purpose of effectively compressing and reconstructing it. The second model is the diffusion model of potential space, which primarily serves to facilitate the transfer of timbre. In order to accomplish domain transformation, the model includes a cross-attention mechanism that makes it easier for information to move from the conditional mechanism to the de-noising UNet. In the end, the conditional mechanism is utilized to obtain the information from the CQT spectrogram of the target instrument and transfer it to the latent space, providing vital data for the entire timbre transfer procedure.

### Perceptual compression

This work builds on the work of *Rombach et al. (2022)* by integrating the idea of perceptual compression to improve the efficacy of training the diffusion model for generating high-quality CQT spectrogram. This feature enhances the computational efficiency of the diffusion model by conducting sampling in a low-dimensional space.

This article employs a convolutional VAE to formally encode the CQT spectrogram $X \in R^{T \times F}$ into a potential space $Z$. Here, $Z \in R^{C \times \frac{T}{r} \times \frac{F}{r}}$, $T$, and $F$ represent the time and frequency dimensions, $C$ denotes the number of channels, and r signifies the compression level of the potential space. In order to achieve high computational efficiency and sample quality, the values of $C$ and $r$ are set to 8 and 4, respectively. Both the encoder $E$ and the decoder $D$ consist of stacked convolution modules, with each block comprising convolution layers and residual connections. During the generation phase, the decoder is utilized to reconstruct $Z$ in order to produce a CQT spectrogram $\tilde{X} \in R^{T \times F}$ for a given potential representation.

In order to train VAE, this study presents the creation of three loss functions: Gaussian constraint loss, adversarial loss, and CQT spectrogram reconstruction loss. The CQT spectrogram reconstruction loss is employed for computing the average discrepancy between the input sample X and the reconstructed CQT spectrogram. The PatchGAN (*Demir & Unal, 2018*) discriminator was employed to enhance the quality of reconstruction in the context of adversarial loss. Gaussian constraints are utilized to impose structure on the potential spaces of VAE, promoting the learning of continuous and organized potential spaces that can better capture the underlying structure of the data. In summary, the general training objectives of VAE can be delineated as follows:

$$\mathscr{L}_{VAE} = \mathscr{L}_{rec}(x, D(\varepsilon(x))) + \lambda\mathscr{L}_{adv}(\psi D(\varepsilon(x))) + \gamma KL_{Gau}(\mu; \sigma^2) \tag{1}$$

where $\mathscr{L}_{VAE}$ denotes the reconstruction loss, $\mathscr{L}_{rec}$ represents the adversarial loss, $\lambda\mathscr{L}_{adv}$ stands for the Gaussian constraint loss, $\psi$ signifies the discriminator utilized in the adversarial process, and $\mu$ and $\sigma$ denote the mean and variance of the VAE potential space.

### Latent diffusion models

This article utilizes a methodology akin to the Palette (*Saharia et al., 2022*) image-to-image conversion technology to instruct the LDM as a timbre transfer decoder. In the initial phase, an efficient and low-dimensional latent space was successfully acquired through the perceptual compression model, wherein high frequencies and subtle details are abstracted. This has a significant impact on the extraction of musical attributes such as pitch, loudness, and timbre. In the subsequent sections, the features extracted from the perceptual compression model will be utilized as input for the constructed LDM model to represent the CQT spectrogram within the potential space and to facilitate the transformation between domain X and domain Y. Upon reflection of LDM, it can be broadly stated that LDM operates by acquiring the ability to produce data from noise through two distinct processes. The initial stage involves the forward process, during which Gaussian noise $\gamma \sim N(0, 1)$ is incrementally introduced to the input until the two become indistinguishable. The subsequent stage involves the reverse process, also known as the de-noising process, during which the decoder acquires the ability to undo the forward process and reconstruct the data from the noise. LDM represents an enhanced iteration of DDPM, sharing the same training process as DDPM. However, distinctions arise in the modeling process. The training process for LDM occurs in a potential space, enabling faster reasoning time.

Given a CQT spectrogram X with a compressed potential encoding $Z_0 \sim q(Z_0)$. In the forward process, starting with the initial data $Z_0$, the diffusion time can be represented as $t \in \{0, 1, \dots T - 1\}$ when $T$ steps are considered. The given code is subject to the incremental introduction of Gaussian noise in accordance with the predefined variance table $\beta_1, \beta_2, \dots, \beta_T$. Following the iterative step $T$, a series of potential noise variables, denoted as $Z_1, Z_2, \dots, Z_T$, is subsequently generated

$$q(Z_{1:T}|Z_0) := \prod_{t=1}^{T} q(Z_t|Z_{t-1}) \tag{2}$$

$$q(Z_t|Z_{t-1}) := \mathcal{N}(Z_t; \sqrt{1-\beta_t}Z_{t-1}.\beta_t I) \tag{3}$$

where $Z_T \in N(Z_T; 0, I)$ is pure Gaussian noise.

In the inverse procedure, commencing with the Gaussian noise distribution $Z_T$ and the desired instrument CQT spectrogram contained within $E^y$, the denoising process, conditioned on $E^y$, progressively produces the potential encoding $Z_0$ of the target CQT spectrogram through the subsequent steps

$$p(Z_t|Z_{t-1}, E^y) := \mathcal{N}(Z_{t-1}; \mu_\theta(Z_T, t, E^y), \sigma_t^2 I) \tag{4}$$

$$p_\theta(Z_{0:T}|E^y) = p(Z_T) \prod_{t=1}^{T} p_\theta(Z_{t-1}|Z_t, E^y) \tag{5}$$

where $\theta$ represents a parameterized neural network defined by Markov chains. In this context, we utilize U-Net, a commonly used model in image synthesis. In practice, the mean $\mu_\theta$ and variance $\sigma_t^2$ are parameterized as follows

$$\mu_\theta(Z_T, t, E^y) = \frac{1}{\sqrt{\alpha_t}}\left(Z_t - \frac{\beta_t}{\sqrt{1-\bar{\alpha}_t}}\varepsilon_\theta(Z_t, t, E^y)\right) \tag{6}$$

$$\sigma_t^2 = \frac{1-\bar{\alpha}_{t-1}}{1-\bar{\alpha}_t}\beta_t \tag{7}$$

where $\alpha_t = 1 - \beta_t$, $\bar{\alpha}_t = \prod_{i=1}^{t} \alpha_i$, and $\varepsilon_\theta(Z_{t,t})$ are predicted generated noise.

In this article, a reweighted noise training objective is used in practice

$$\mathcal{L}_{simple}(\theta) = \mathbb{E}_{\varepsilon(x),\varepsilon\sim\mathcal{N}(,),\ell}||\varepsilon_\theta(Z_t, t, E^y - \varepsilon)||_2^2 \tag{8}$$

where $\varepsilon \sim \mathcal{N}(0, I)$ is derived from a diagonal Gaussian distribution.

The style transfer module is utilized for modeling the CQT spectrogram in potential space and executing complete style transfer, as depicted in Fig. 3. Given a source audio sample CQT spectrogram $X$, its potential representation $Z_{t_0}$ is calculated by adding step $t_0 < T$ noise according to Formula (3). Subsequently, commencing with $Z_{t_0}$ as the initial stage of the reverse process, the conditional mechanism is employed to access the potential encoding of the target instrument CQT spectrogram $Y$

$$p_\theta(Z_{0:t_0}|E^y) = p(Z_{t_0}) \prod_{t=1}^{T} p_\theta(Z_{t-1}|Z_t, E^y) \tag{9}$$

Therefore, in order to accomplish the transformation of style. If $t_0$ controls the outcomes, the original audio information will not be preserved in the case of $t_0 \approx T$.

This research employs a $2 \times 2$ convolution layer, which is a two-dimensional convolution layer, in experiments to enhance the capabilities of the underlying U-Net. To improve the U-Net backbone, a cross-attention mechanism is incorporated so that it can produce CQT spectrogram of the target domain according to the requirements needed to

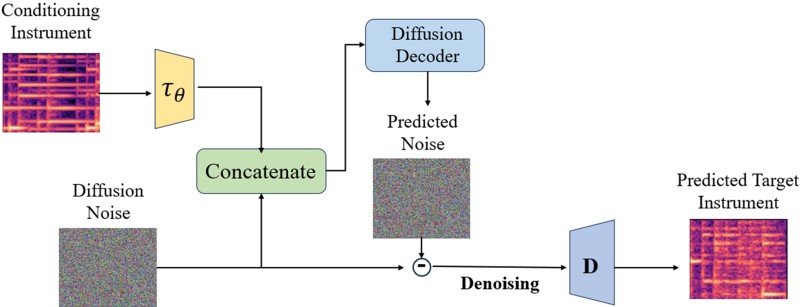

**Figure 3 Timbre style transfer architecture.**

accomplish timbre transfer. Three subsampling blocks, each with four filters and residual blocks of 64, 128, and 256, make up the encoder in U-Net. To condense the possible encoding of the CQT spectrogram, average pooling with a pooling size of two is applied after each subsample block's output. A cross-self-attention block follows the encoder's last block. The bottleneck part obtained by the encoder is processed by a residual block with 512 filters and then passed to the decoder. The only difference between the symmetric decoder and encoder is that the decoder uses transposed convolution to create an upsampled layer that increases the feature dimension. The cross-self-attention layer succeeds the last subsampling block, the encoder's bottleneck, and the decoder's first higher sampling layer.

*Conditioning mechanisms*

The conditional mechanism primarily comprises encoder $\tau_\theta$, which is structured in a manner similar to the encoder used in perceptual compression. The encoder $\tau_\theta$ projects the input target instrument CQT spectrogram, represented by $y$, into an intermediate representation $\tau_\theta(y) \in \mathbb{R}^{M \times d_\tau}$ after encoding it. The cross-attention layer is then used to map this intermediate representation to the middle layer of the U-Net, making it easier to create a CQT diagram that depicts the target instrument's timbre under condition $y$. The formula for the cross-attention mechanism is expressed as follows

$$Attention(Q, K, V) = softmax\left(\frac{QK^T}{\sqrt{\bar{d}}}\right) \cdot V \tag{10}$$

where $Q = W_Q^{(i)} \cdot \varphi_i(Z_t)$, $K = W_K^{(i)} \cdot \tau_\theta(y)$, $V = W_V^{(i)} \cdot \tau_\theta(y)$. $\varphi_i(Z_t) \in \mathbb{R}^{N \times d_\varepsilon^i}$ represents the U-Net intermediate representation that implements $\varepsilon_\theta$. $W_V^{(i)} \in \mathbb{R}^{d \times d_\varepsilon^i}, W_K^{(i)} \in \mathbb{R}^{d \times d_\tau}$ and $W_Q^{(i)} \in \mathbb{R}^{d \times d_\tau}$ are projection matrices that map intermediate representations from $\tau_\theta(y)$ to the target domain, thereby enabling the transfer of timbre. The objective function can be rewritten as

$$\mathscr{L}_{LDM}(\theta) = \mathbb{E}_{\varepsilon(x),y,\varepsilon \sim N(0,1)}||\varepsilon - \varepsilon_\theta(Z_t, t, \tau_\theta(y))||_2^2 \tag{11}$$

where $\tau_\theta$ and $\varepsilon_\theta$ can be optimized by the objective function.

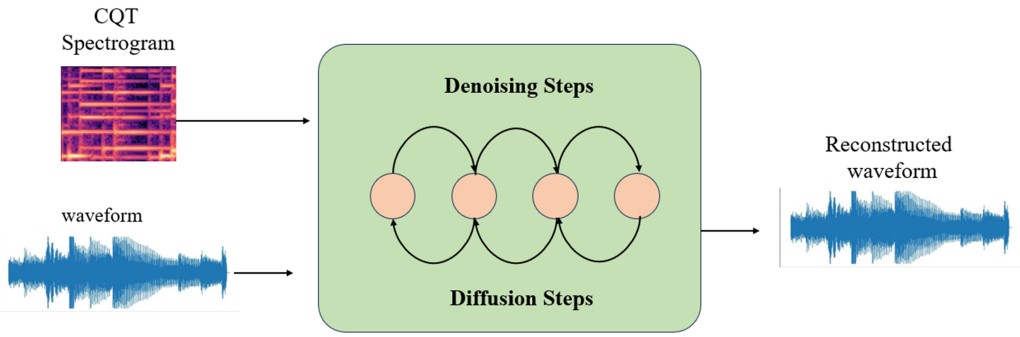

**Figure 4 DiffWave architecture.**

## Waveform reconstruction

In the preceding section, we successfully analyzed the input CQT spectrogram and acquired the CQT spectrogram generated after the timbre transfer by following the processes outlined. Converting the CQT spectrogram into an audio waveform is challenging due to the absence of phase information. This restriction can be overcome by creating waveforms using neural networks. Numerous neural vocoders, including WaveNet (*Oord et al., 2016*; *Rethage, Pons & Serra, 2018*), WaveGAN (*Yamamoto, Song & Kim, 2020*), MelGAN (*Jain et al., 2020*), and other models, are composed of neural networks. Among the various models, DiffWave is notable for its compact size and capacity to rapidly produce high-quality speech waveforms. However, limited research has been conducted on its capability to generate music audio (*Kandpal, Nieto & Jin, 2022*).

DiffWave is a neural vocoder and waveform synthesizer that operates on a diffusion model. The process commences with Gaussian noise and subsequently transforms it into speech through iterative refinement. The generation of speech can be controlled by providing a conditional signal, such as a log-scale Mel-spectrogram. Diffwave comprises a sequence of residual layers featuring a biaxially expanded convolution structure. A schematic representation of the DiffWaves model is depicted in Fig. 4. In order to produce this, the characteristics of the input are upsampled to match the dimensions of the anticipated waveform. Subsequently, the transition distribution in the inverse process is systematically sampled to acquire the waveform. In this study, the generation process is dependent on the CQT representation.

As proposed in *Jain et al. (2020)*, the objectives of minimization in the training process are defined as follows

$$\min_{\theta} \mathcal{L}_{unweighted}(\theta) = \mathbb{E}_{x_0,\varepsilon,t} ||\varepsilon - \varepsilon_{\theta}(\sqrt{\bar{\alpha}_t}x_0 + \sqrt{1 - \bar{\alpha}_t}\varepsilon, t)||_2^2 \quad (12)$$

where $t \in \{0, 1, ..., T - 1\}$, $\varepsilon_{\theta}$ represents a neural network. For more comprehensive information, readers are encouraged to refer to the citations.

## EXPERIMENTS

This section outlines the experiments conducted to validate the performance of the proposed timbre transfer technique in both single-instrument and multi-instrument application scenarios.

## Experimental setting

### Description of the data

We used the MusicNet and MAESTRO datasets, together with the real-world recordings from other instruments that we gathered from YouTube, to train and assess the model described in this research. The dataset comprises instruments such as piano, flute, guitar, clarinet, violin, trumpet, organ, strings, and vibraphone. Each instrument collected approximately 2 h of data, during which the tracks underwent quasi-switching to mono and resampling to 16 kHz. The WAV format was used for preprocessing on all data sets. The data set was split up in an 8:1:1 ratio between a training set, a test set, and a validation set.

### Implementation details

In this study, the Adam optimizer with a batch size of 16 was employed to train the proposed timbre transfer model over 50,000 iterations, with a learning rate set at 2e-5. The compression ratio of the potential space is 64. During the timbre transfer process, the denoising size was configured to 0.55, the seed size to 42, the steps per sample to 50, and the guidance to 7. The DiffWave model trained over 20,000 iterations at an initial learning rate of 2e-5 using the CQT spectrogram condition and an Adam optimizer with a batch size of 32. It is important to note that all models developed in this study are built upon the Pytorch library. All models were trained using thre NVIDA RTX3090Ti graphics processing units.

### Evalutaion metrics

The performance of the proposed model is assessed using both objective evaluation and subjective evaluation methods in this study.

In the objective evaluation, the following indicators were adopted:

- **Jaccard distance:** The Jaccard distance is utilized to quantify the disparity between two sets of pitches, A and B, in order to evaluate the extent to which the produced tracks vary in pitch profile. The Jaccard distance can be computed using the following formula:

$$JD(A,B) = 1 - |\frac{A \cap B}{A \cup B}| \tag{13}$$

  The Jaccard distance values range from 0 to 1, where lower values indicate fewer mismatches, indicating a greater degree of similarity between the generated pitch profiles. The pitch profile is determined through the utilization of the multi-pitch version of MELODIA (*Salamon & Gómez, 2012*) incorporated in the Essentia library (*Bogdanov et al., 2013*), with the pitch being rounded to the nearest semitone.

- **SSIM:** The SSIM (Structural Similarity Index) (*Setiadi, 2021*) is employed for assessing the perceived quality disparity between the original image and its reconstructed counterpart. Here, the reconstructed CQT spectrogram are compared with the original spectrogram to evaluate the fidelity of the model reconstruction. The SSIM value ranges from −1 to 1, with a higher value indicating greater structural similarity between images.

- **Fréchet Audio Distance (FAD):** FAD (*Roblek et al., 2019*) is utilized to quantify the similarity between synthesized audio and authentic audio. Based on the PyTorch implementation, the FAD was calculated by embedding the VGGish model (*Shi et al., 2019*). The approach considers the embeddings as a continuous multivariate Gaussian distribution and computes the Frechet distance between the real and generated data. The formula for calculating FAD is as follows:

$$FAD = ||\mu_r - \mu_g||^2 + tr\left(\sum_r + \mu_g - 2\sqrt{\sum_r \sum_g}\right) \tag{14}$$

where $(\mu_r, \sum_r)$ represents the embedded mean and covariance of the real data, and $(\mu_g, \sum_g)$ represents those of the generated data. A smaller FAD value results in a more realistic sample set being generated.

- **Accuracy:** We have developed multiple instrument classifiers through training. A network resembling AlexNet (*Iandola et al., 2017*) was trained using audio clips extracted from a dataset to categorize the transmitted audio based on different instruments. The output of the classifier undergoes processing by the Sigmoid function, which yields the classification probability for each segment. In this article, the classification probability is articulated as a measure of confidence.

The mean opinion score (MOS) was utilized for subjective evaluation. The MOS score was calculated based on an anonymous listening test involving 50 participants. The scoring system ranges from 1 (low) to 5 (high) and encompasses three dimensions:

1. Success in style transfer (ST): evaluating the degree to which the timbre of the transfer version aligns with the target in perception.

2. Content retention (CP): the degree to which the transmitted version of the music content matches the original version.

3. Sound quality (SQ): How does the audio sound as a whole?

## Experimental analysis

### Evaluation of timbre transfer

This section will assess the impact of timbre transfer. As demonstrated in the prior study (*Lu et al., 2019*), our objective was to assess our methodology using three criteria: (1) content retention, which measures the degree to which the tonal content of the input is preserved in the output; and (2) style fit, which evaluates the extent to which the output aligns with the target timbre. (3) The quality of the audio. In this regard, we employed both subjective and objective indicators to assess the measurements.

**Subjective evaluation.** To subjectively evaluate the ability of timbre transfer, the MOS of the listening test was gathered from 50 participants. The participants in the study primarily consisted of individuals with a strong affinity for music. The target mixture was utilized as the reference standard for each work in each evaluation cycle, and the appropriate translated output was given after that. The conditions and examples were presented in a random order in each segment, which the tester was not aware of

**Table 1 Five-scale MOS of one-to-one timbre transfer.**

| Task | ST | CP | SQ |
|------|------|------|------|
| Piano to Guitar | 4.02 | 3.95 | 4.12 |
| Flute to Trumpet | 3.92 | 4.04 | 4.06 |
| Trumpet to Piano | 4.05 | 4.10 | 4.15 |
| Flute to Organ | 3.96 | 4.01 | 4.14 |

beforehand. The test is segmented into two parts, involving one-to-one and many-to-many conversions.

The initial stage of the assessment involved a one-on-one instrument conversion segment, during which the participant listened to eight musical pieces representing four types of transitions: piano to guitar, flute to trumpet, trumpet to piano, and flute to organ. In each round of scoring, the order of conditions and examples in each individual part of the test was randomized. The results obtained through the hearing test in the first stage are presented in Table 1.

The results indicate that our model demonstrates strong performance in the task of one-to-one instrument timbre transfer. The MOS for both style transfer and content retention are around four, indicating that our model performs well on these tests. It has been noted that our approach works well in cases where the target instrument has a relatively "stable" sound, such as in the conversion of a piano. Meanwhile, the sound quality scores for all one-to-one conversion tasks are high, indicating that our method performs well in generating audio, although the generated audio is quite a bit different from real-world recordings.

In the subsequent phase, we assessed the task of one-to-many timbre style transfer. In the context of one-to-many timbre style transfer, this study has chosen six instruments from the test set and conducted transfers between each instrument and the other five instruments, as well as the original instrument, resulting in a total of 36 transfer pairs ($6 \times 6 = 36$). Simultaneously, we also took into account four different tracks. Two of the tracks pertain to the conversion from vibraphone/clarinet to piano/violin, while the other two are related to the reverse conversion. The conditions and examples in the test were randomized, and the order was not known to the participants beforehand. Also, all participants rated the output in isolation. The MOS score is presented in the three dimensions of ST, CP, and SQ and subsequently averaged to obtain the final assessment score. A heat map was generated to illustrate the hearing test outcomes of one-to-many transfer for a single timbre, as depicted in Fig. 5. Table 2 presents the auditory test results of multi-pair multi-transfer for mixed instruments.

Differences in performance can be observed among various timbre transfers in the context of many-to-many transfers of a single timbre. Additionally, it is evident that the timbre reconstruction effect surpasses the transfer effect. The transfer of timbre from other instruments to the piano is notably effective, while the transfer to the trumpet is comparatively less successful. One possible explanation is that the timbre of the piano

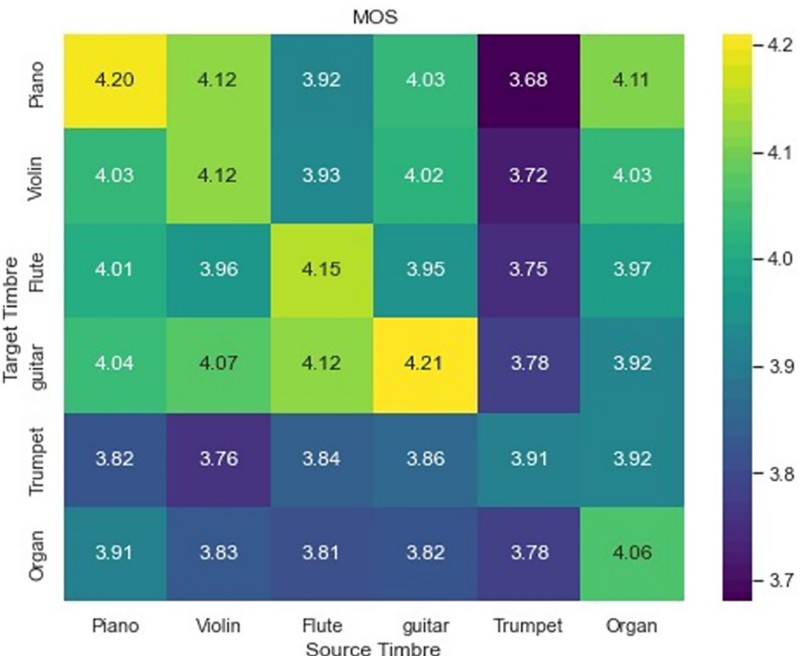

**Figure 5 MOS for one-to-many timbre transfer outcomes.**

**Table 2 MOS of multi-timbre transfer.**

| Task | ST | CP | SQ |
|------|-----|-----|-----|
| Vibraphone/Clarinet to Piano/Violin | 3.68 | 3.71 | 3.91 |
| Piano/Violin to Vibraphone/Clarinet | 3.71 | 3.69 | 3.92 |

changes more regularly compared to the trumpet, because the decay of the piano timbre after each note is struck is relatively predictable. This relative regularity of timbral characteristics makes piano timbral characteristics easier to learn in timbre space. Conversely, the trumpet's timbre exhibits such significant variation that it is challenging to master and manipulate. The data presented in the table indicates that our approach is comparatively less effective in facilitating many-to-many conversion of multi-instruments as opposed to single-instrument many-to-many conversion. This disparity may be attributed to the challenges associated with extracting and distinguishing the timbre of mixed instruments.

**Objective evaluation.** In the objective evaluation, this study will employ the Jaccard distance to assess the content retention ability of the model, and FAD to examine the perceived similarity between the generated audio and the original audio. The SSIM is used to measure the perceived similarity between the reconstructed CQT spectrogram and the original spectrogram in order to evaluate the model's ability to generate a high-quality CQT spectrogram. An instrument classifier is utilized to assess the precision of the model's timbre transfer. The Mel-frequency cepstrum coefficient (MFCC) is widely regarded as a

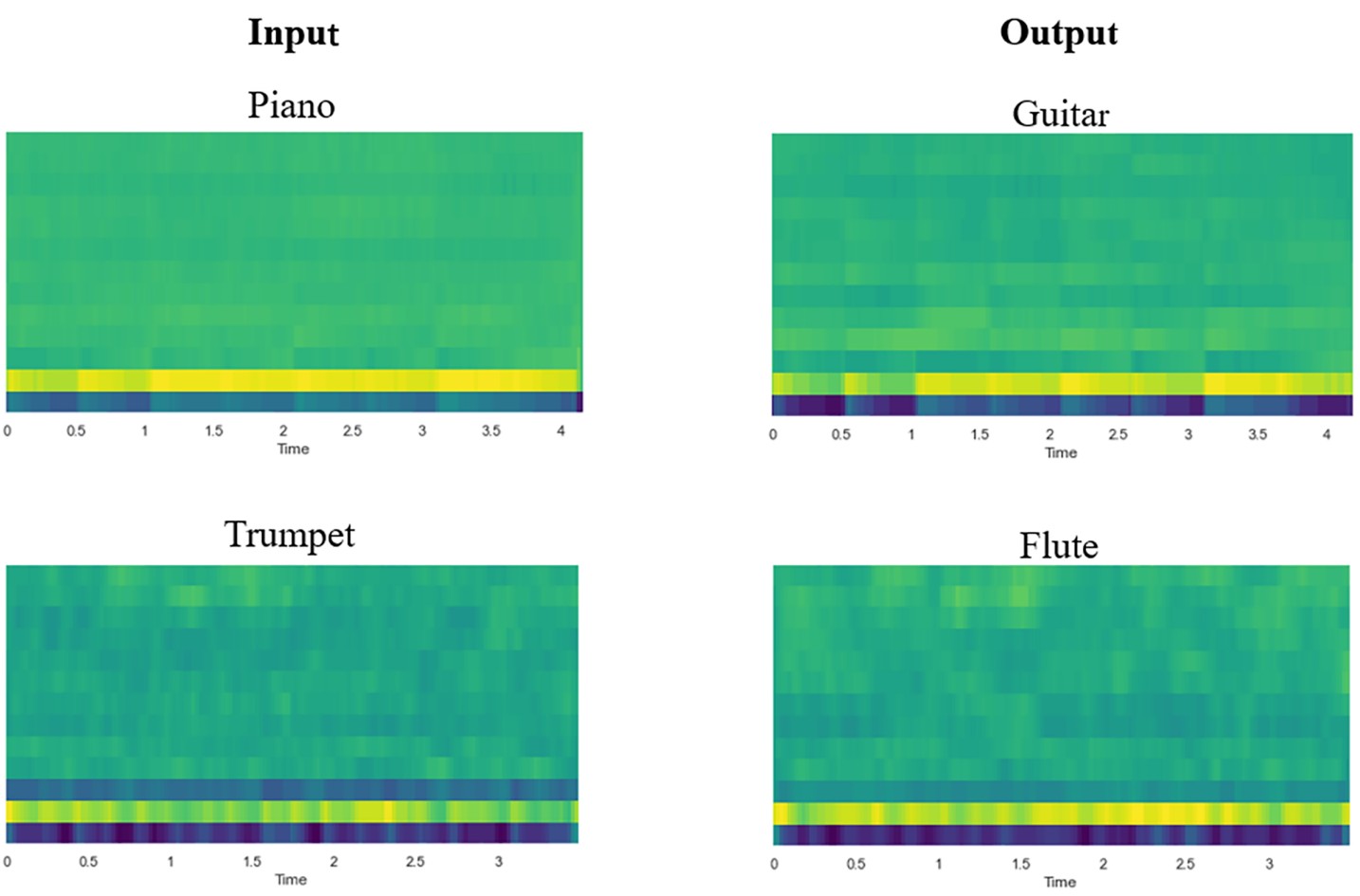

**Figure 6 Diagram of Mel-frequency cepstral coefficients (MFCC) for input and output audio.**

reliable approximation of timbre (*Richard, Sundaram & Narayanan, 2013*). To facilitate a more intuitive observation of the timbre transfer of the instrument, Fig. 6 displays the MFCC diagram of several sets of input and output audio. The results of the objective evaluation are presented in Table 3.

The timbre transfer between the input and output audio may be clearly seen in the picture. Consistent with the subjective evaluation, the objective evaluation statistics show that one-to-one conversions for single instruments perform better than many-to-many conversions for mixed instruments. From the data, it can be observed that the model proposed in this article has high values in terms of accuracy, indicating that the model performs well in the forced-choice classification task. Although high accuracy means that the output sound is closer to the timbre of the target instrument, accuracy alone does not provide a comprehensive measure of the quality of timbre transfer, and further sonic assessment and subjective evaluation are necessary. It is also noteworthy that the generated pitch contour closely resembles the pitch contour of the target input, as indicated by the low JD values. This illustrates how well the content was retained between the input and the

**Table 3 Findings of an objective assessment.**

| Task | FAD↓ | Accuracy↑ | SSIM↑ | JD↑ |
|---|---|---|---|---|
| Piano to Guitar | 3.16 | 98.5% | 0.82 | 0.32 |
| Flute to Trumpet | 4.12 | 95.7% | 0.84 | 0.39 |
| Trumpet to Piano | 4.25 | 94.1% | 0.79 | 0.43 |
| Flute to organ | 3.23 | 98.2% | 0.88 | 0.30 |
| Vibraphone/Clarinet to Piano/Violin | 6.56 | 91.1% | 0.77 | 0.48 |
| Piano/Violin to Vibraphone/Vlarinet | 7.05 | 90.6% | 0.78 | 0.49 |

generated audio. Higher SSIM values indicate that the CQT spectrograms generated by the model are of good quality, which is one reason why subsequent music generation has a lower FAD. Concurrently, the low FAD value indicates that, as expected, the audio output of the model has better perceptual quality.

The analysis of the objective results largely corresponds to the subjective reviews, which is as we expected. This suggests that our model is capable of handling both one-to-one and many-to-many timbre transfer tasks. This confirms that the diffusion model performs well on timbre transfer and music synthesis tasks, offering researchers a more intriguing direction for their work. However, in contrast to the one-to-one style transfer task, the many-to-many task was relatively poor, suggesting that the model is underperforming for learning and extracting timbres from mixed instruments. This suggests that the model struggles to accurately separate and learn the timbres of mixed instruments, which is a direction that needs to be pursued in the future.

### Additional properties

Apart from the primary assessment outcomes, we investigated several aspects of the model, specifically the compromises between sampling step and quality, as well as the relationship between compression ratio and quality. Figure 7 displays the outcomes of our trials, which involved the piano2guitar and vibraphone/clarinet2piano/violin tasks. These tasks represent one-to-one and many-to-many timbre transfer tasks, respectively.

**Trade-off between sampling steps and quality.** From the results in the Fig. 7, it can be noticed that increasing the number of sampling steps in the potential diffusion model improves the quality, which may be due to the fact that the generated potential space is more detailed and at the same time produces better overall structured music. However, at 50-100 sample steps, there is no significant decrease in the FAD values with increasing sample steps, which may be related to the fast fitting of the model. Considering the speed relationship, we weighed the selection of 50 as the sampling step. This is because model generation becomes slower as the number of sampling steps increases.

**Trade-off between compression ratio and quality.** From the results in Fig. 7, we can see that lowering the compression ratio improves the quality of the generated music, because a low compression ratio is closer to the original data, but it also slows down the running speed of the model. Considering that we need to process higher dimensional data later, we weigh the performance of the model and choose a compression ratio of 64.

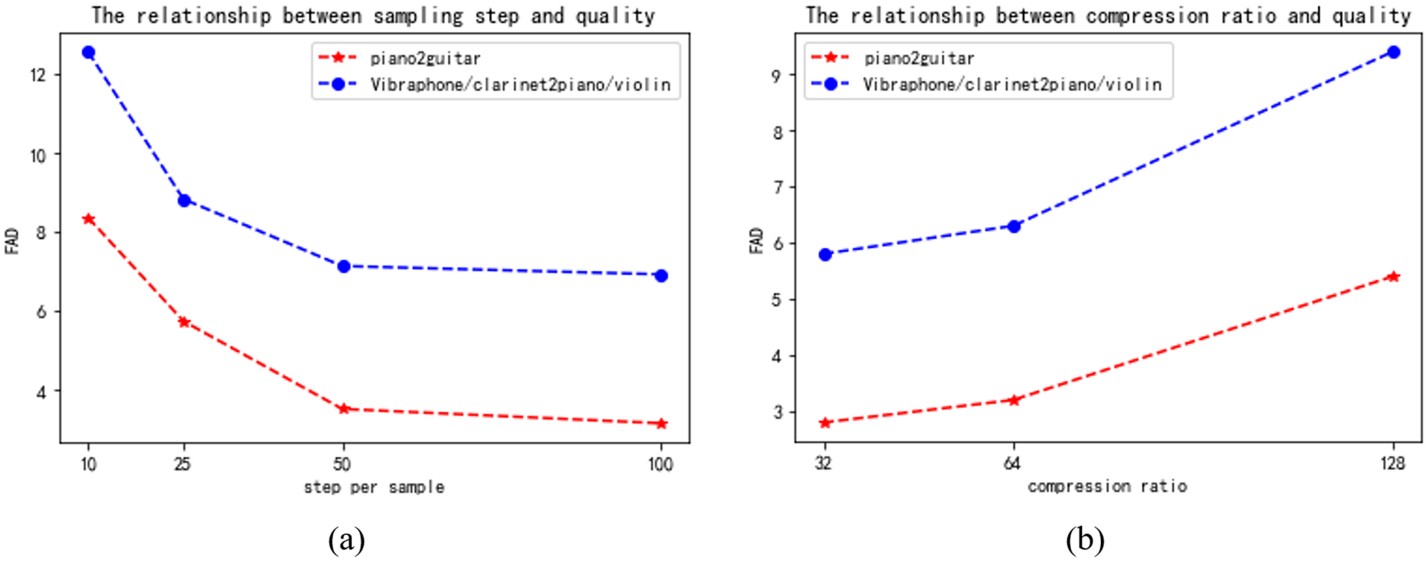

**Figure 7** (A) The relationship between sampling step and quality. (B) The relationship between compression ratio and quality.

### Comparison with baseline model

To validate the effectiveness of the method proposed in this article, we consider a comparison with three baseline models. For the baseline models, we consider VAE-GAN (*Bonnici, Benning & Saitis, 2022*), Music-Star (*Alinoori & Tzerpos, 2022*) and DiffTransfer (*Comanducci, Antonacci & Sarti, 2023*) in this article. The VAE-GAN architecture combines a variational encoder and a generative adversarial network for constructing meaningful representations of source audio and generating realistic target audio. It enables many-to-many timbre style transfer, where the dataset of this article is used for training according to the procedure described in *Bonnici, Benning & Saitis (2022)*. Music-Star is an audio translation system based on the WaveNet autoencoder, which is capable of converting audio waveforms into the styles of different musical instruments. The model used in this article was trained based on the description found in the literature (*Alinoori & Tzerpos, 2022*). DiffTransfer is a timbre transfer method for both single and multi-instrument use that is based on a denoising diffusion implicit model. In this study, the model is trained using the description of *Comanducci, Antonacci & Sarti (2023)*. The findings of an objective and subjective assessment of the proposed method in comparison to the baseline model are given. We consider piano-to-guitar and piano-to-vibraphone timbre transfers, for one-to-one instrument transfers. For many-to-many transformations involving audio from multiple instruments, we consider timbre transfer from vibraphone/clarinet to piano/strings. Table 4 presents the outcomes of the subjective assessment. In every section of the test, the conditions and examples were presented in a random order, and the participants were unaware of the method used to generate them. Table 5 and Fig. 8 show the outcomes of the objective assessment, where the results in Fig. 8 are presented to verify the accuracy of the proposed model with respect to the success of the style transfer

**Table 4** MOS with the baseline comparison model.

| Model | Task | | | | | | | | |
|---|---|---|---|---|---|---|---|---|---|
| | Piano to Guitar | | | Piano to Vibraphone | | | Vibraphone/Clarinet to Piano/Strings | | |
| | ST | CP | SQ | ST | CP | SQ | ST | CP | SQ |
| VAE-GAN | 3.96 ± 0.09 | 3.84 ± 0.10 | 3.82 ± 0.09 | 3.78 ± 0.08 | 3.82 ± 0.10 | 3.75 ± 0.09 | 3.65 ± 0.11 | 3.56 ± 0.08 | 3.52 ± 0.09 |
| Music-Star | 4.02 ± 0.10 | 3.92 ± 0.11 | 3.85 ± 0.08 | 3.92 ± 0.06 | 3.95 ± 0.08 | 3.94 ± 0.07 | 3.56 ± 0.10 | 3.54 ± 0.11 | 3.61 ± 0.11 |
| DiffTransfer | 4.08 ± 0.09 | 4.01 ± 0.08 | 4.10 ± 0.07 | 4.01± 0.06 | 4.03 ± 0.08 | 4.12 ± 0.10 | 3.80 ± 0.08 | 3.71 ± 0.07 | 3.84 ± 0.10 |
| ours | 4.12 ± 0.10 | 4.05 ± 0.10 | 4.13 ± 0.11 | 4.02 ± 0.07 | 4.06 ± 0.08 | 4.15 ± 0.08 | 3.78 ± 0.06 | 3.69 ± 0.08 | 3.92 ± 0.09 |

**Table 5** Results of the objective evaluation contrasted with baseline models.

| Model | Task | | | | | |
|---|---|---|---|---|---|---|
| | Piano to Guitar | | Piano to Vibraphone | | Vibraphone/Clarinet to Piano/Strings | |
| | FAD | JD | FAD | JD | FAD | JD |
| VAE-GAN | 8.41 | 0.54 | 9.16 | 0.56 | 12.52 | 0.67 |
| Music-Star | 6.47 | 0.39 | 7.43 | 0.41 | 10.93 | 0.57 |
| DiffTransfer | 3.34 | 0.31 | 4.56 | 0.28 | 6.73 | 0.46 |
| DiffTransfer (DiffWave) | 3.20 | 0.30 | 4.31 | 0.28 | 6.43 | 0.47 |
| ours | 3.16 | 0.32 | 4.22 | 0.29 | 6.37 | 0.48 |

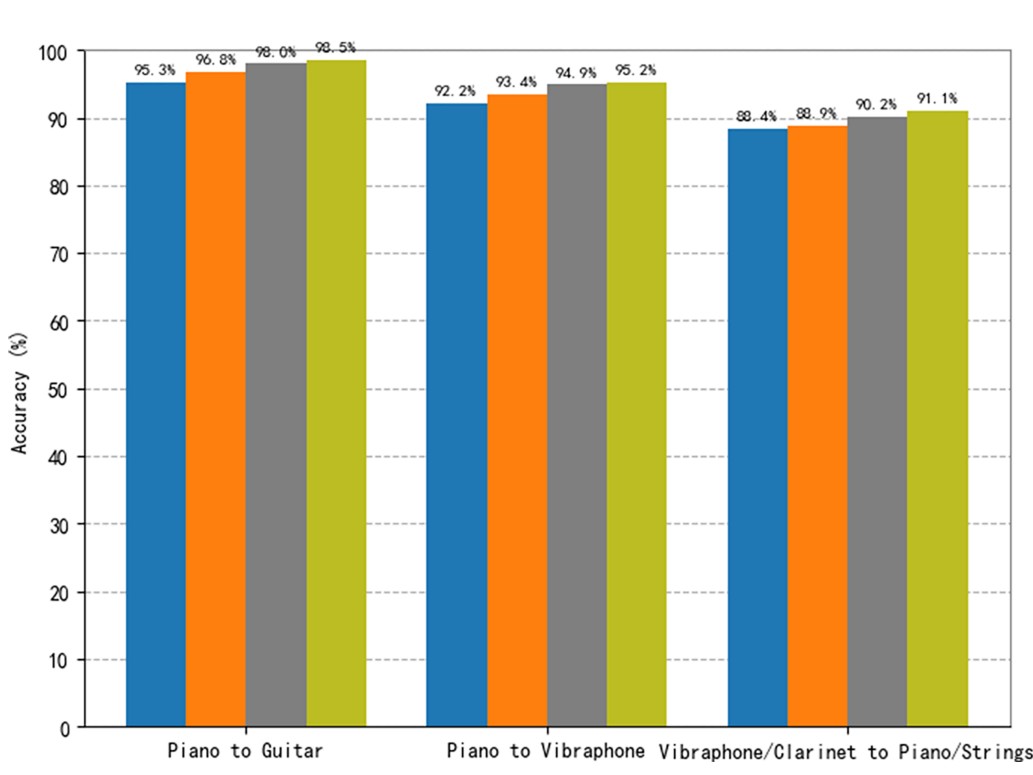

**Figure 8** Accuracy with the baseline comparison model.

from the baseline model. The classification criteria use the same instrument classifiers trained on a network similar to AlexNet (*Shi et al., 2019*).

The suggested approach performs better than the two baseline models, VAE-GAN and Music-Star, according to the subjective scores in Table 4. On the other hand, in comparison to the DiffTransfer model, the performance is comparable. This demonstrates the favorable performance of our method. We note that our method shows good sound quality relative to the baseline model, suggesting that DiffWave is a promising direction for the field of music synthesis.

The classification results presented in Fig. 8 demonstrate that the audio output from the approach suggested in this research performs much better in terms of resemblance to the target instrument timbre than both VAE-GAN and Music-Star. Regarding DiffTransfer, the classification outcomes show a slight gain over the baseline model, but the difference is not significant. This demonstrates that within the topic of music style transfer, diffusion modeling is a promising direction. The technique suggested in this article performs much better in constructing single and multiple instrument timbre transfer tasks than the two baseline models, VAE-GAN and Music-Star, as can be seen by a brief review of the data provided in Table 5. The results also demonstrate that, in terms of FAD values, the approach suggested in this research marginally outperforms the DiffTransfer baseline model; this result may be explained by the usage of the DiffWave vocoder model in this work. Nevertheless, the approach suggested in this article is somewhat less than the DiffTransfer baseline model when the JD value is taken into account. This could be because the latent diffusion model used in this article necessitates compression in the potential space.

Consider that the choice of vocoder may have an impact on the performance of timbre transfer. For a more comprehensive comparison, we used the DiffWave vocoder with the original SoundStream vocoder in our experiments with the baseline model DiffTransfer. The results show a slight performance advantage of DiffWave over SoundStream in terms of sound quality. Specifically, in the Piano2Guitar task, the DiffWave version of DiffTransfer had a FAD value of 3.20, which was slightly better than the SoundStream version of DiffTransfer at 3.34. In the Piano2Vibraphone and Vibraphone/Clarinet2Piano/Strings tasks, the performance is also slightly better. This shows that DiffWave is a promising direction in the field of music synthesis.

Upon conducting a concise analysis of both subjective and objective outcomes, it is evident that the proposed approach demonstrates favorable performance in the transfer of timbre. Compared to the baseline model, the diffusion model performs better in the timbre transfer task. This illustrates how effectively diffusion models perform in music synthesis and timbre transfer tasks when compared to models like VAEs and GANs, and it offers an appealing option for subsequent research.

## CONCLUSIONS

In this research, we offer a diffusion model-based flexible-timbre transfer method. This technique enable the timbre transfer of both single and multiple instruments. This study provides potential layers to minimize the dimension of the data, therefore accelerating the

model's inference. A cross-attention mechanism is concurrently added to the potential layer to learn the target instrument timbre from the conditional mechanism, thereby achieving the timbre transfer. This work uses the CQT spectrogram approach for the time-frequency representation of audio. Lastly, the CQT spectrogram created after conversion is restored using the DiffWave model. Through objective evaluation and subjective hearing tests, we compare the proposed method with the baseline model; the experimental results indicate that the new method performs better. We want to adjust the conditional approach in the future to enable the model to learn and implement the transfer of genres and composition styles.

### Funding
This research is funded by the Science and Technology Innovation Team in College of Hunan Province (nos. Xiang Jiao Tong 2023-233) and the Natural Science Foundation of Hunan Province (nos. 2022JJ50002, 2024JJ7091). The funders had no role in study design, data collection and analysis, decision to publish, or preparation of the manuscript.

### Grant Disclosures
The following grant information was disclosed by the authors:
Science and Technology Innovation Team in College of Hunan Province: 2023-233.
Natural Science Foundation of Hunan Province: 2022JJ50002, 2024JJ7091.

### Competing Interests
The authors declare that they have no competing interests.

### Author Contributions
- Hong Huang conceived and designed the experiments, performed the experiments, analyzed the data, performed the computation work, prepared figures and/or tables, authored or reviewed drafts of the article, and approved the final draft.
- Junfeng Man conceived and designed the experiments, authored or reviewed drafts of the article, and approved the final draft.
- Luyao Li analyzed the data, prepared figures and/or tables, and approved the final draft.
- Rongke Zeng analyzed the data, prepared figures and/or tables, and approved the final draft.

### Data Availability
The demo is available at Youtube: https://youtu.be/DTINzPH_LBI.
The code is available at GitHub and Zenodo:
- https://github.com/AlexHuang44/timbre-transfer-with-diffusion-modelcode
- Huang, H. (2024). Timbre transfer. Zenodo. https://doi.org/10.5281/zenodo.10811176
The MAESTRO Dataset is available at: https://magenta.tensorflow.org/datasets/maestro.

MusicNet data is available at Zenodo: John Thickstun, Zaid Harchaoui, & Sham M. Kakade. (2016).

MusicNet (1.0) [Data set]. Zenodo. https://doi.org/10.5281/zenodo.5120004.

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
