# Peer review of "Musical timbre style transfer with diffusion model"

_PeerJ Computer Science, doi:10.7717/peerj-cs.2194_

## Round 0.1 · original submission · Major Revisions

Both reviewers raise very different types of concern with the submitted paper. I would expect a resubmitted paper to contain major changes in the writing - to account for the relationship of this work with previous epochs of research, with other latest results raised by reviewer 2, and reviewer 1's concerns about the quality of outputs and the actual needs of the envisioned use.

**Language Note:** The review process has identified that the English language must be improved. PeerJ can provide language editing services - please contact us at [email protected] for pricing (be sure to provide your manuscript number and title). Alternatively, you should make your own arrangements to improve the language quality and provide details in your response letter. – PeerJ Staff

Reviewer 1 ·

Basic reporting

In terms of basic reporting, the article is clearly written and the experimentation seems sound, but there are still major problems that must be addressed.

This manuscript presents an extremely naive understanding of music and while this is common in the field, it is not acceptable. Multiple claims in the introduction are flawed and ignore decades of work in computer music and perhaps centuries of music theory. I will try to be concrete in the following.

"Prior to exploring this matter, it is generally accepted that music is made up of two complimentary components: style and content." This is far from accepted. Read any music text and see if you see this claim. Maybe it is true that "Researchers interested in machine-learning approaches to music processing have found it convenient to make a distinction between style and content." but that is quite different.

"While the definition describes what timbre is..." -- no, the definition clearly describes what timbre is *not*. This has been pointed out often in the literature.

"... modeling timbre becomes a challenging task due to significant differences in the time and frequency domains among different instruments." -- this is only one of many problems.

The first reference for timbre transform is 2018. What about cross-synthesis? E.g. Burred, "Cross-Synthesis Based on Spectrogram Factorization," ICMC 2013, or Lazzarini and Timoney, "New Methods of Formant Analysis-Synthesis for Musical Applications", ICMC 2009, or Serra, "Sound Hybridization Based on a Deterministic Plus Stochastic Decomposition Model", ICMC 1994. Admittedly, many of these consider the combination of different aspects of sounds from different sources and not always an attempt to strictly separate pitch and loudness from timbre, but one could also point to MIDI as exactly a representation of pitch and loudness, with "timbre" as an independent variable. You might say this is synthesis only, but consider MIDI as an intermediate representation between analysis and synthesis, e.g. guitar synthesizers by Roland for "timbre transfer from electric guitar" or the IVL PitchRider (around in the 1980s) for real-time audio-to-midi which could drive synthesizers. Also work at MTG at UPF driving SIS synthesis from real-time audio input. You might also say these are not learning-based systems. But of course the output sounds are captured or modeled in some way, so I would say you need to define things carefully. It's safer and more intellectually honest to view machine learning approaches as a new approach to a problem that composers have investigated for decades.

"previous methods have complex structures," - but again, you have ignored most previous methods, only going back to 2018.

"Here, we define pitch, loudness, and duration as the fundamental characteristics of music, while considering the timbre of instruments as the element that contributes to its style." Normally, you can define things any way you want, but I think this is really going out on a limb. This suggests that the element that changes between swing and samba is timbre, while pitch, loudness and duration are invariant.

You also have not talked yet about time scales. When you say "loudness" do you mean the overall loudness of a tone, expressed as a single quantity, or are you talking about the evolution of loudness over time? Same for pitch. It is an important distinction. E.g. Dannenberg's Introduction to Computer Music (Lulu .com) emphasizes that loudness and pitch control are essential to "musical results" (e.g. realistic timbre), so if you attempt to "transfer" time-varying loudness, you *cannot* "transfer" timbre at high-quality.

Or else, you have to carefully define what you mean by timbre. Saying that it "contributes to its style" is not very meaningful.

There are many references to achieving "high-quality audio" but if you think the cited results have high-quality, you should talk to musicians or producers or recording engineers. Except in the area of vocal synthesis, where there is either more experience in the speech area to draw on, or perhaps the natural jitter and noise in speech production masks the problems, the general consensus is that these deep-learning synthesis methods are "low quality" and that the authors have very low standards. Things are changing all the time, but you can listen to the cited works and hear distortion and noise levels that are unacceptable for professional work in music. The provided examples sound terrible. Just compare, for example, the piano used as input to the piano produced by the model. It is fine to claim an interesting technique or progress, but you claim "the proposed model can produce high- quality audio". This is simply false by any reasonable measure of quality. Later, it is revealed that the bandwidth is 8kHz vs. 22kHz bandwidth of CDs, cosidered a minimum for modern music recording.

"RELATED WORD" should be "RELATED WORK" (p4)

"spectrograml" (multiple places) is spelled "spectrogram"

"music pitch follows an exponential pattern rather than a non-linear one." - exponential *is* non-linear. I think you mean linear.

Line 427: "Meir frequency" should be "Mel-frequency"

Experimental design

The paper presents a good description of the methods used and some justification, and the comparison with some previous work as baseline are good, but given the low quality of the results and the apparently even lower quality of the baselines, in my opinion the article would be much more useful as an analysis of what techniques are contributing to improvement (to the extent individual design decisions can be evaluated), what is the sensitivity of the results to various design parameters, and what seems to be limiting the ultimate quality? Obviously, the sample rate is an important factor, and scaling up sample rates seems to be a weakness that should be discussed. However, even given the bandwidth limitation, the sound is far from natural, so the obvious question is not "how did you get 'good' results?" but what is holding these methods back from achieving even moderately good results?

Validity of the findings

At some level, the work is fine and clearly reported.

However, the overall tone is claiming a successful solution when the results are obviously poor and only incremental improvements over prior work. This is an active area, so I am not convinced this new work really represents the state-of-the-art, but at least it is an interesting effort that is described clearly.

As mentioned above, and in the interest of scientific progress, I would recommend to focus on an analysis of the design in the hope of better understanding how to make these systems get better results.

Cite this review as
Anonymous Reviewer (2024) Peer Review #1 of "Musical timbre style transfer with diffusion model (v0.1)". PeerJ Computer Science

Reviewer 2 ·

Basic reporting

see below

Experimental design

see below

Validity of the findings

see below

Additional comments

This paper presents a method of timbre style transfer using a kind of diffusion model which has gained success in image-to-image style transfer. Here, Timbre transfer is achieved by transferring the style information of a CQT image while preserving the content information. In addition, the vocoder part of the proposed system is also based on diffusion model (diffwave). In general, the paper is technically correct, all the details are well described, and the experiment validation is also sufficient.

However, I am not sure whether the authors are aware of this work:

[1] Luca Comanducci, Fabio Antonacci, Augusto Sarti, Timbre Transfer Using Image-to-Image Denoising Diffusion Implicit Models, ISMIR 2023.

It should be noted that the main idea of this under-review paper is almost the same with [1]: using Palette (Saharia et al., 2022) on CQT images. [1] was published in Nov. 2023, which is 2 months earlier than the time this paper is submitted to the Journal. Since the experiment part of the reviewed paper and [1] are different, this case should not be taken as plagiarism. However, it is true that the idea presented in this paper is not new.

Since the experiment part is the only way to gain novelty, my suggestion to the authors is to extend this part. The authors are therfore encourged to 1) add more technical details (e.g., the motivation to present Figure 7 and how the classification was performed? Since your are performing classification, I think you can report the results for all the samples used in your subject test?), 2) provide more in-depth discussion, e.g., why the diffusion model outputs better results and what are the insights gaind from the experiment result?

Demo audio should also be provided.

Other minor comments:

Line 33: Research on music style transfer is broadly categorized into timbre style transfer, performance style transfer, and composition style transfer -> need reference

Cite this review as
Anonymous Reviewer (2024) Peer Review #2 of "Musical timbre style transfer with diffusion model (v0.1)". PeerJ Computer Science

---

## Round 0.2 · Major Revisions

Please see the details of the reviewer's feedback.

Reviewer 1 ·

Basic reporting

The manuscript is much better in terms of putting the work in context that is much broader than recent deep learning models.

The other reviewer pointed out (slightly) prior work in ISMIR 2023. This work should be cited and discussed. The submitted work uses a previous study as a baseline, but given the ISMIR reference, the obvious question now is whether the "baseline" is appropriate. In particular, the claim is improved performance over the baseline, but one has to ask whether the baseline is still a good choice for comparison, and what conclusions can we draw?

It is still claimed that the results are "high quality." If these results are high quality, then the term "high quality" is almost meaningless. Given decades of standards that support >20kHz bandwidth for music audio and lots of research to back up that choice, how can you claim results that are band-limited to 8kHz are "high quality"?

This also brings into question the methodology and interpretation of results, discussed below.

Throughout the article, tasks are described with phrases such as "many-to-many transmission for a single instrument" and "multi-pair multi-transmission for mixed instruments". First, I think you mean "transfer" and not "transmission". Second these terms are ambiguous, e.g. "many" could mean instruments playing simultaneously (polyphony) or it could mean a single model capable of dealing with multiple instruments. "mixed" could mean "various types of" or "playing simultaneously." In the introduction, you should clearly define terminology and various tasks to be studied and modeled.

Experimental design

Any subjective experiment that leads to the conclusion that highly band-limited music audio is "high quality" must be examined carefully. Subjects were asked to rate the audio on a 5 point scale as to sound quality, and the conclusion was that a rating of >4 indicates high quality. However, there is no reference for comparison. What would happen if instead of rating output in isolation, subjects also rated actual instrumental recordings at CD quality and listened to on high-quality headphones? Or what would happen if subjects were given references to calibrate what quality should get ratings from 1 to 5? Or what would happen if subjects were asked to compare the quality of experimental results with a reference and report better, same, or worse?

Similar questions could be asked about objective measures. It seems that the measurements are objective, but there is nothing there to lead one to the conclusion that the results are "high quality" in any absolute sense.

The baseline comparisons are good and meaningful, but the choice of baseline is questionable given more recent (but still prior to submission) work.

Validity of the findings

As mentioned, claims of "high quality" might have some plausibility in this narrow context, as in "high-quality in the context of low expectations and lack of competitive alternatives" but even in the context of consumer electronics (CDs and MP3s for example), the bandwidth alone rules out any hope of high quality music audio.

Additional comments

There are a few typos: line 156 "we presents"
line 201: "this chapter" (section? article?)
line 300: "a 22 convolution layer" - not sure what this means, is it a typo? 2x2?

Cite this review as
Anonymous Reviewer (2024) Peer Review #1 of "Musical timbre style transfer with diffusion model (v0.2)". PeerJ Computer Science

---

## Round 0.3 · Major Revisions

There is an important translation error. In your new manuscript, you refer to "migration" many times ("timbre migration", "style migration"), and this is the wrong word. Please change all occurrences of "migration" back to "transfer", and then re-submit the document (and the other documents).

---

## Round 0.4 · Minor Revisions

The article has been improved, thank you. However, reviewers raise issues especially with clarity of the writing. PeerJ criteria state that articles "must use clear, unambiguous, technically correct text" and this is important. Please address the listed issues. Technical clarity is important.

Reviewer 1 ·

Basic reporting

It was suggested to state the problem clearly. New lines 97-102 attempt to do this, but the writing is still not clear. First, you are creating a function that maps from input to output, so describe both input and output. "The first involves transferring just the timbre of a single instrument in a one-to-one fashion." What do you mean by "transfer"? Is content involved? What do you mean by "one-to-one" - one what transfers to one what? Each note? Each instrument (but you said there's only one instrument)? In line 92, you call this "shifting between tracks", but it seems odd to describe any digital signal processing in this way.

"The second task is known as the many-to-many transfer of a single instrument and involves the flexible transfer of audio waveforms comprising only one instrument between several instruments." is also unclear. I think you are making a distinction between monophonic or single-voice (but "voice" does not seem to be a term that non-musicians understand) or melody and polyphonic or multiple-voice music, all comprised of a single class of instrument. Note that "single instrument" is confusing throughout the paper because it could mean one instance of an instrument or multiple simultaneous notes played by multiple instruments, but restricted so that all instruments are of the same type. Note that piano, vibraphone and even violin are capable of playing more than one note at a time. Is a chord played on piano or violin an example of the first problem (because it's a "single instrument") or the second (because it is polyphonic)?

The third task refers to "mixed instruments" and "transfer across the mixed tracks" but it's unclear what tracks have to do with anything, and it is odd to say "mixed instruments". In fact, I think the concept you are after is really single timbre vs multiple timbre instead of single instrument vs multiple instrument (and note that these could be sounding simultaneously or not -- I think you are always assuming simultaneous, but this is never stated). I suppose the problem is, given a source with multiple timbres, to preserve the pitch, duration and amplitude properties of all sounds, while replacing all first input timbres with the first output timbre, all sounds with the second input timbre are replaced with the second output timbre, etc.

Line 454: "transfer of skills" is completely unclear; is it a typo?

Experimental design

no comment

Validity of the findings

Once again, there are unsupported claims of sound quality:
Line 433 "our model can effectively replicate the target timbre." No, your data suggests your model can achieve certain mean opinion scores in some tests, but there is nothing to relate the scores to the quality or degree if timbre replication.

Line 437 "our method has good sound quality." No, there is no attempt to define "good sound quality" or relate it to the ratings.

Line 475 "model in this work has a high degree of accuracy" indicating "timbre transfer ... was successful" - since accuracy refers to a forced-choice classifier, this is a pretty low bar. Effectively, you are saying the work is successful if the output sounds more like a guitar than a few other possibilities. But this view of effectiveness was not stated as a goal of the project, nor should it be.

Line 483 "model can perform reasonably well" - this is not saying much, but even then there's no definition of what is expected or reasonable. To me, the sounds are of unusable musical quality, so I would not say the model performs reasonably well.

Table 4 does not show any standard deviations and the results do not include a significance test. It seems that there are very small differences being treated as significance, e.g. Line 538 "demonstrates superior performance compared to the baseline."

Line 575: "can produce high-quality audio" - again, there was no expert opinion, no comparison to sounds of actual acoustic instruments, no calibration of mean opinion scores, so there is no reason to believe or conclude the resulting audio is of high quality (and obviously to my ear, it is not).

Additional comments

You refer to "piano's stable sound" (line 456). I think you mean the relative stability of the spectrum of the piano compared to trumpet. Piano sound is certainly unstable with respect to it's rapid decay from the onset.

Line 458: "trumpet's tone" - is this also a reference to the spectrum? Tone is not a clear term.

Line 520: "enables musical reprise of audio waveforms" - is this another attempt to use a new word, similar to the earlier use of "shifting between tracks"? "Reprise" is not a good word here.

Line 579: "so" is not a good word choice. Maybe "thereby" or "thus" but probably better to just word things differently.

Cite this review as
Anonymous Reviewer (2024) Peer Review #1 of "Musical timbre style transfer with diffusion model (v0.4)". PeerJ Computer Science

Reviewer 2 ·

Basic reporting

See below

Experimental design

See below

Validity of the findings

See below

Additional comments

The revised manuscript has included a new baseline, which is the ISMIR 2023 paper I mentioned last time. Now the presentation of the paper makes more sense. I still have the following few comments and strongly encourage the authors to revise them:

1. Describe the difference between your work and the ISMIR 2023 paper in the introduction. I only see such discussion in the experiment result section (and the discussion is still weak, see below). Or, the introduction section would read like there is no contribution in this paper.
2. On the discussion at line 564: if the vocoder can affect the performance, could you also try SoundStream (the vocoder used in the ISMIR 2023 paper) on your model (or DiffWave for their model) for better comparison?
3. The author should double check the format of the reference. Now the formats of the references are quite inconsistent. For example, the paper (Comanducci et al.) was published in 2023 but its year attribute is 2024. Also, the conference and journal names need to be made consistent.

Cite this review as
Anonymous Reviewer (2024) Peer Review #2 of "Musical timbre style transfer with diffusion model (v0.4)". PeerJ Computer Science

---

## Round 0.5 · accepted · Accept

The authors have made good attempts to satisfy the reviewers' requests for clearer technical language. In my judgment the improved text is now ready for publication.